# The Spaces of Peer-Led Support Groups for Suicide Bereaved in Denmark and the Republic of Ireland: A Focus Group Study

**DOI:** 10.3390/ijerph19169898

**Published:** 2022-08-11

**Authors:** Lisbeth Hybholt, Agnes Higgins, Niels Buus, Lene Lauge Berring, Terry Connolly, Annette Erlangsen, Jean Morrissey

**Affiliations:** 1Psychiatric Research Unit, Psychiatry Region Zealand, 4200 Slagelse, Denmark; 2Mental Health Services East, Psychiatry Region Zealand, 4000 Roskilde, Denmark; 3School of Nursing and Midwifery, Trinity College Dublin, D02 PN40 Dublin, Ireland; 4School of Nursing and Midwifery, Faculty of Medicine, Nursing and Health Sciences, Monash University, Clayton, VIC 3800, Australia; 5Department of Regional Health Research, University of Southern Denmark, 5000 Odense, Denmark; 6Friends of Suicide Loss (FOSL), D06 T685 Dublin, Ireland; 7Danish Research Institute for Suicide Prevention, Mental Health Centre Copenhagen, 2900 Hellerup, Denmark; 8Copenhagen Research Centre for Mental Health, 2900 Hellerup, Denmark; 9Department of Mental Health, Johns Hopkins Bloomberg School of Public Health, Baltimore, MD 21205, USA; 10Center of Mental Health Research, Australian National University, Canberra, ACT 0200, Australia

**Keywords:** peer-support, postvention, suicide bereavement, qualitative research

## Abstract

Research has shown that people bereaved by suicide have an increased risk of mental health problems, suicidality and associated stigma, as well as higher rates of sick leave and increased rates of receiving disability pensions. Peer-led suicide bereavement support groups are perceived to enhance people’s recovery by enabling shared experiences with others who are bereaved in similar circumstances. The aim of the research was to explore the viewpoints of participants living in Denmark and the Republic of Ireland on these peer-led support groups. This study focused on how the participants experienced being part of the peer-led support and how the participation affected them. We conducted four focus groups, two in Denmark and two in the Republic of Ireland, and two individual interviews, involving a total of 27 people bereaved by suicide. Data were analyzed thematically. The participants’ experiences in the peer-led support groups were in contrast to what they had experienced in their daily lives. They felt alienated in daily living, as they believed that people could not comprehend their situation, which in turn led participants to search for people with similar experiences and join the peer-led support groups. While peer-led support groups may not be helpful for everyone, they created ‘supportive spaces’ that potentially affected the participants’ recovery processes, from which we generated three key themes: (i) ‘A transformative space’, describing how the peer-led support group created a place to embrace change, learning and knowledge about suicide and suicide bereavement and the making of new connections; (ii) ‘An alternative space for belonging and grieving’, describing how the participants felt allowed to and could give themselves permission to grieve; and (iii) ‘A conflicted space’ describing how it was troublesome to belong to and participate in the peer-led support groups. In conclusion, despite the two cultural settings and different organizational approaches, the experiences were comparable. Peer-led support groups can, despite being a conflicted space for some, provide supportive spaces aiding the participants’ recovery process.

## 1. Introduction

The impact of death by suicide is extensive, and it has been described as a death like no other [1]. Many people struggle in the aftermath of suicide and are at an increased risk of prolonged grief [2]. Death by suicide is associated with troublesome emotions and feelings such as blame, guilt, emptiness and shame [3,4]. Grief after suicide can be considered a ‘disenfranchised grief’ [5], as the grieving person is often unable to publicly mourn the loss because of a societal reluctance to discuss death by suicide, which can increase people’s feeling of isolation.

The World Health Organization [6] estimates that worldwide approximately 700,000 people die by suicide annually, with Andriessen et al.’s [7] meta-analysis suggesting that 1 in 20 people are affected by suicide in any given year and that 1 in 5 are affected during their lifetime. An American survey found that 51% of 1432 participants had experienced one or more suicides by someone they knew during their lifetime, out of which 35% indicated having suffered from moderate to severe emotional distress after the loss [8]. Being bereaved by suicide has been linked to social withdrawal, reduced psychological and somatic functioning, and an increased prevalence of mental health morbidities such as depression, anxiety disorders and psychiatric hospitalization [9,10,11]. In addition, bereaved relatives also have an elevated risk of suicide themselves when compared to those bereaved by homicide or other types of sudden death [12]. Prevalence rates of 38% and 8% have been reported for suicidal ideation and suicide attempt, respectively, among people bereaved by suicide [13]. Higher rates of sick leave and becoming the recipient of a disability pension have also been shown among those bereaved by suicide [14].

According to the dual-process model of coping with bereavement [15], navigating the bereavement process is a combination of an inner psychological experience and a social and cultural event. Bereavement may change a person’s identity as they work on their new roles, relationships and circumstances, which may include a continuing bond with the deceased [15]. A Japanese study found that people bereaved by suicide conceptualize the aftermath as a process that moves from experiencing life as out of their control to living a redefined life. Two types of people were identified in the study: (i) ‘dialoguers’, who receive comfort through open dialogue, and (ii) ‘mood changers’, who tended to conceal their emotions and suffering [16]. Thus, it seems that the needs for support are likely to depend on the individual’s way of dealing with the added challenges associated with suicide bereavement. Andriessen [17] (p. 43) defined postvention as ‘activities that have been developed to facilitate recovery after suicide, and to prevent adverse outcomes including suicidal behavior’. One such activity is peer support groups, which are sometimes called mutual-support groups. Although definitions of peer support vary, peer support usually refers to support that is reciprocal in nature and occurs between people who share similar life experiences [18]. In addition to providing support, empathy, compassion, validation and a sense of belonging and community, peer groups are perceived to provide social spaces, which enables people to negotiate new understandings and meanings of their distress, allowing them to become active participants in their own recovery [19,20].

Mental health policies increasingly dictate that support services should engage with and involve people with lived experience in the design and delivery of services and interventions [18]. Studies that focus on peer-led suicide bereavement interventions are scarce. In a systematic review of peer-led postvention, Higgins et al. [21] identified 10 papers of low to moderate quality. Findings from this review suggested that peer-led suicide bereavement interventions potentially might improve participants’ well-being, sense of connectedness, hope and grief by normalizing the loss, reducing self-blame, isolation and stigma. Nevertheless, some participants expressed dissatisfaction with the intervention or reported distress from listening to others’ grief. Suicide bereavement support groups initiated by people with lived experiences are likely to be the primary resource for many bereaved people, and considering the limited evidence in this area, there is a need for more research [22,23]. Qualitative research can contribute towards important findings regarding participants’ experiences of postvention, for instance, what is perceived as helpful or not [24]. In this study, we focused on the perceived usefulness of those peer-led groups in two countries, i.e., the Republic of Ireland and Denmark. The analogue assessment allowed for a comparison of similarities and differences across cultural settings, thus potentially being able to induce some universal features related to the experiences of participating in a peer-led support group.

### Aim of This Study

The aim of this study was to explore participants’ perspectives on peer-led support groups for people bereaved by suicide. In this paper, we focused on how the participants experienced being part of a peer-led support group in Denmark and the Republic of Ireland and how the peer-led support affected them.

## 2. Method

### 2.1. Study Design

We applied a qualitative descriptive design where focus groups and individual interviews were conducted to collect data for a reflexive thematic analysis. A qualitative descriptive design was selected to address first-person perspectives of people with lived experiences and provide rich descriptions of their experiences [25]. Focus groups were chosen as they were considered the preferred method to facilitate the participants to share their experiences of being a member of a peer-led support group, and to provide insight into the participants’ experiences and processes as a group member [26]. To enhance participation, individual interviews were offered to those who wished to participate but could not attend the focus group. A growing body of literature points towards the importance of patient and public involvement in the design and conduct of research [27]. The chairperson of the Irish organization ‘Friends of Suicide Loss’ initiated this research project, and representatives from the two organizations involved in the study provided feedback on the design and the formulation of the interview schedule.

### 2.2. Setting for the Postvention Services

Both the ‘National Association for the Bereaved by Suicide’ (NABS) in Denmark and the Irish organization ‘Friends of Suicide Loss’ (FOSL) were initiated by people bereaved by suicide, in Denmark in 2002 [28] and in the Republic of Ireland in 2015 [29]. The philosophy behind both organizations is underpinned by the belief that if people bereaved by suicide engage in dialogue with peers, it can support their grieving process. At the same time, those bereaved by suicide gain inspiration to move forward in their lives in a constructive way. Both organizations are underpinned by ground rules centering on confidentiality, respect and a non-judgmental approach. As both groups are not-for-profit organizations, they are supported economically through donations, fundraising and, in Denmark, by a small membership fee.

NABS provides a range of activities such as café evenings (open groups), where people meet, talk and share their experiences, peer-support via the telephone, walk and talk, access to books on relevant topics and weekend seminars/workshops with a focus on sharing experiences and learning how to help oneself. While people bereaved by suicide lead the work, professionals are invited to contribute to seminars/workshops. All activities are advertised in newsletters, on social media platforms such as Facebook, and on their website. Everybody can join in all activities without invitation or prior appointment, except for the seminars that need registration and a small payment.

FOSL provides a similar range of activities, such as coffee mornings, walk and talk, workshops, symposiums, conferences, hosting of ‘International Survivors of Suicide Loss Day’, access to a library on suicide bereavement for members, support via telephone, email and online platforms, outreach support to families, if needed, in their own homes and peer-led support group meetings. The core group, mainly long-term members, meets twice monthly for a period of two hours, and people are free to join, leave and re-join the group on an ongoing basis. The newly bereaved groups meet weekly and participate in a 6–10-week program. To join an FOSL group, the individual first contacts the named group ‘organizer/facilitator’, who meets with the person to discuss their expectations, provides information about the group format and ground rules, and jointly decides if the peer group is suitable for the person at that time.

### 2.3. Recruitment

Participant recruitment was based on purposive sampling focused on recruiting ‘information-rich’ participants [30], rather than a statistically representative sample. Inclusion criteria specified that participants had to (i) be over the age of 18, (ii) be bereaved by suicide within more than one year and (iii) have participated in peer-led support groups conducted by NABS in Demark or the Irish organization FOSL. Both organizations were involved in advertising the study through posts on social media. Potential participants contacted the research team directly or gave permission to the respective organizations to pass on their contact details. Following this, a member of the research team provided detailed written and oral information and subsequently answered any questions. Once all participants had their queries addressed, a date and time for the focus group was set.

### 2.4. Participants

In total, 27 people participated, 14 in Denmark and 13 in Ireland, with a gender mix of 8 men and 19 women. The participants’ kinship to the deceased person varied and included children, siblings, partners and spouses (See Table 1). Five participants reported experiencing multiple losses by suicides. The duration of time since the bereavement varied from 1 to 44 years, with the majority being bereaved within the previous 10 years. Some participants had met through the activities in the organizations and thus knew each other before the focus groups. Members of the same family were allocated to separate focus groups. The participants had typically attended several of the activities described above, including support groups and self-help seminars/workshops. The Danish participants had attended NABS for a period ranging from 1 to 19 years and the Irish participants for a period from 1 to 5 years.

### 2.5. Focus Groups

In Denmark, two focus groups [26] were conducted face to face in hotel facilities. Due to COVID-19 restrictions in the Republic of Ireland at the time, two focus groups were conducted online on Microsoft Teams. The length of the focus groups was between 120 and 150 min. During the focus group, one of the facilitators made field notes of the interactions in the group. In Ireland, individual interviews were conducted via telephone (approx. 60 min each) with two people who were unable to participate in the online focus groups.

In Denmark, a senior researcher in psychiatric nursing (L.H.) and a research assistant facilitated the focus groups. In the Republic of Ireland, an assistant professor (J.M.) and a professor in mental health nursing (A.H.) conducted the focus groups. With the exception of the research assistant, all were Ph.D.-educated female researchers and from disciplinary backgrounds of mental health nursing, counselling and with extensive experience in qualitative research. Only the participants and facilitators were present. The researchers did not know the participants personally. All (both face to face and online) focus groups were audio-recorded, as well as the interviews. Data were collected in 2021.

The focus groups were guided by a schedule (see Appendix A) developed by the researchers in consultation with representative members from the two organizations, NABS and FOSL. The goal was to stimulate dialogue and group interaction [26], and the interview schedule was mainly used as a reminder of relevant topics to address rather than as a protocol to be followed. The focus groups were opened with a brief round of introductions and a short input by the moderators on the format, including information on how data were stored, analyzed and published to protect confidentiality. All focus groups ended with a thank you and a reminder of the available support, including a follow-up telephone call from a research team member. The focus groups in Denmark were conducted in Danish, and professional translators were used to translate the interview schedule and the selected quotations used in this article. A Danish-speaking researcher (L.H.) reviewed all translations.

Overall, the conversation in the focus groups flowed freely, with the moderators only occasionally needing to engage to seek clarification. The way in which the participants described their experiences of participating in the peer-led support group was reflected in the focus groups, with people listening attentively to each other and ensuring everyone had space to speak uninterrupted. Participants sometimes assisted one another, finding words, and gently reminded each other if someone was going off track. Further, they mainly spoke from a personal or ‘I-position’, i.e., saying ‘this is how it is for me’, while being mindful that it may not be the same for others. When differences in opinion or experience were expressed, they did not judge others but used it as an opportunity to reflect on their own views and experiences.

### 2.6. Data Analysis

The focus groups and interviews were transcribed verbatim and analyzed using a reflexive thematic approach inspired by Braun and Clarke [31,32,33]. Reflexive thematic analysis was used to generate patterns of shared meaning across a qualitative dataset through a reflective and thoughtful engagement with the data [33]. The process of analysis was iterative, moving back and forth between reading and coding of the data, comparing codes across groups and countries, combined with dialogue among the research team to agree on generated themes. Over time, this emphasis changed as researchers explored their understanding of the data corpus and compared data across groups and countries. At the outset, the focus was on how participation in the peer-led support groups influenced recovery in the aftermath of the suicide loss. Later, the analytical emphasis was on the parts of the data corpus that focused on how the participants described the peer-led support groups as different spaces compared to other settings in their everyday life and how peer-led support affected them and their daily living.

Braun and Clarke’s 6 phases [31] of analysis were merged into 4 phases according to the needs of our study, as detailed below.

#### 2.6.1. Phase 1: Familiarizing Yourself with Your Data

The first phase involved letting members of the research team listen to the audio recordings, reading and re-reading the transcripts from their own country. LH, NB and LLB focused on the Danish transcriptions, and JM and AH revisited the Irish transcriptions. The emphasis was on sharing initial impressions and identifying potential meanings, patterns and codes. Once this was completed, both groups connected to discuss their developing ideas.

#### 2.6.2. Phase 2: Generating Initial Coding

Data were coded in the temporal order of events, commencing with the experiences of suicide bereavement through to their experiences of the peer-led support groups. This was followed by the exploration of the participants’ thinking and emotions in relation to the peer-led support groups and their impact. First, each group summarized its country’s findings; then the groups came together to combine their findings and discuss differences and similarities across the two countries.

#### 2.6.3. Phase 3: Focused Coding

In this phase, the focus was on the individual and collective activities/processes that had taken place in the peer-led support groups. Emphasis was placed on how people described and interpreted the activities, as well as what flowed from the activities within and outside the group. The coded data from each country were compared and summarized and formed the basis for the next dialogue in the research group.

#### 2.6.4. Phase 4: Theme Identification

As an outcome of the dialogue, potential themes were discussed. At this stage, it was agreed to write memos on each theme and include selected quotations to illustrate the themes. Once completed, the research team once again scrutinized and discussed the similarities and differences emerging from the two cultural contexts and agreed on the final title of themes, including the representative quotations that exemplified them.

### 2.7. Ethical Considerations

In accordance with Danish legislation, the Danish Data Protection Agency (Reg-117-2020) and the regional research ethics committee (J.nr. 20-000013) were notified about the Danish arm of the study. The Irish arm of the study received ethical approval from the Faculty of Health Science Research Ethics Committee (FHSREC), Trinity College Dublin. The study adhered to the ethical principles of the Declaration of Helsinki [34]. Participation was informed and voluntary. Written consent was obtained from all participants. To ensure participants’ well-being, a follow-up telephone call was made after the interviews. There were no reports of any negative effects. Due to data protection issues, the sharing of raw data across national borders was not permitted. The data are not publicly available due to the rules of the ethical committee. To protect the participants’ confidentiality, we use pseudonyms in the quotations.

## 3. Results

Participants from both cultural settings, i.e., Denmark and the Republic of Ireland, described how their motivation for seeking peer-led support came from their encounters with people in their network and with professionals. They described how their social network had focused on the deceased person and on comforting them in the first period after the suicide. However, as time passed, they felt that being bereaved by suicide limited their sense of social connectedness and alienated them from others. They perceived a lack of understanding for their loss, emotions, reactions, situation and grief. They experienced a difference between ‘us’, who were bereaved by suicide, and ‘others’, who were not able to comprehend the depth of their pain. This affected their regular social interactions and resulted in different social sanctions, such as people avoiding them and trying to close down any conversation about grief or trying to find simple solutions. In essence, they perceived that people expected them to get over their grief and move on with their lives, which made them feel uncomfortable expressing their emotions.

Most of the participants sought support from different health professionals, professional-led support groups or religious groups. There were mixed views about the support from these sources. While some acknowledged how the professionals’ theoretical insight helped them understand their own and their relatives’ grief processes, others did not find the professional support helpful. For example, Amine stated ‘*I went to see a counsellor, I only went twice because she just didn’t understand what I was going through*’ (Irish focus group 1). They all longed for community and a sense of togetherness, which they did not find in encounters with laypeople or professionals; hence, they searched for peers to help them address their grief and to handle the aftermath of suicide. The following three themes were generated from the thematic analysis, which described the experiences of the bereaved in the peer-led support group and how it affected them: (i) ‘A transformative space’, (ii) ‘An alternative space for belonging and grieving’ and (iii) ‘A conflicted space’. The themes were identified in both cultural settings. We use the term space, as it was a term used by the participants, which not only represented the physical reality that existed in the support group, but it also captured the interpersonal space that participants created and held for each other. We found that this space existed equally via Zoom meetings as well as face-to-face and as such became a prominent concept in our analysis.

### 3.1. A Transformative Space

The peer-led support group was perceived as a transformative space because it enabled participants to process the change and find new ways to come to terms with their grief. It also helped them handle the expectations and reactions of people in their social network ‘outside’ the support group. For example, Alexis describes how she often had to take on a role in daily living, pretending she was doing fine, but felt devastated inside:


*Alexis: I think I would be able to get an Oscar statuette for best acting performance. Because you go on and live and manage your life. You manage your life and you play some character, but in the peer group the masks drop completely. They are not needed at all among peers.*
(Danish focus group 2)

The peer-support group was described as an empowering space where participants could disclose and display their grief, which helped them handle societal expectations in their daily lives. They learned to understand their grief by listening to and comparing themselves with the narratives of peers, for example, how they dealt with needs and challenges in the aftermath of suicide. They learnt not only how to process their grief by re-telling their own stories, reading books and listening to podcasts recommended by the group, but also how to keep the memory of the deceased alive without being overwhelmed by it. The peer-led support group enabled the participants to learn about themselves, their situation and how they could regain a meaningful life with the bereavement as a lifelong companion. In the following quotation, Mary described how she recognized her own feelings when listening to the stories of peers who enabled her to believe and see recovery as a possibility:


*Mary: The main thing I found was that here were other people who were at different stages, some were further down the road, and they were surviving. Because I couldn’t figure out how to survive I didn’t know how long this freaking out was going to last. So, when I went to the group, other people were managing it, they were back in their jobs doing various things, and as I listened and heard their stories, I realized that they were feeling the same things I was feeling. And they were managing and it kind of gave me courage and hope.*
(Irish focus group 1)

The mutual sharing activated new reflections and insights, which broadened their understanding of themselves and others. By comparing themselves with their peers, they identified similarities and differences in situations, reactions, feelings and actions. Those who told their story processed their grief by narrating their own situation and by listening to the narration of others; this initiated reflections that influenced how they handled and understood their reactions and situation in daily living. The participants described how every time they shared experiences in the group, it became a little easier to bear the grief in general. For example, in this dialogue, the participants talk about how being in the support group prompted reflection and action:


*Kimberly: […] Participation in activities affects you.*



*Clare: Yes, it affects in the sense that it starts things in yourself in terms of what you are going through. You hearing others describe how they deal with matters or have read something or the like, which may be just one small sentence that starts something in you. Then you begin contemplating ‘Well, yeah, that is one way to deal with it—perhaps I should try and find out something’.*



*Taylor: Some people are better at wording their feelings. I mean something has been obscure to me, and then I listen to someone who just states it so clearly.*



*Clare: Like they hold the missing piece of the puzzle.*
(Danish focus group 2)

Listening to others telling and re-telling their stories of their loss validated their own experiences. In addition, telling and retelling of their own story taught them to be more empathetic to themselves and others and gave them a sense of hope and strength to live with the loss. The support and learning from the peer-led support group empowered participants to share with people outside the group, thus reducing the feeling of alienation, even though they would not be able to fully understand the bereavement process:


*Helen: I was so devastated, but then coming back 2 or 3 years later I am so much stronger. I have told my story over and over again in the group and then one day where I was, I was able to talk openly and I didn’t mind if I cried. I was able to. And because the way I could handle it they appreciated as well. You know they had compassion with me as well. But it was the group that, it was like a family behind you, you had the strength, do you understand?*
(Irish focus group 2)

The retelling over and over of their story gave them the opportunity and ability to find the language to express their inner world; as the story was retold over and over, it began to change, and they were able to find the words and the strength to express themselves. The support group community also had the potential to reach outside the group setting, as peers from the support group offered support and comfort, for example, by sending a text message on anniversaries or an open invitation to make a phone call or take a walk when needed. Some participants described how, as a result, they had established a network of peers who became part of their private daily lives. The nature and impact of the support provided is captured in Rowan’s narrative:


*Rowan: I have received a lot of help besides the group. Several peers said I could always call or suggested we can do this and that, which I have used a lot. For example, my husband jumped from a bridge, and I did not ever, never want to get near it, but then a peer said let’s go for a walk to see the bridge. She literarily held my hand when we went to look at what turned out to be a complete harmless bridge… and it was such a relief. I do not know what I had imagined; that there would be a huge sign, but it was just a bridge. I have received a lot of help from the members outside the group sessions.*
(Danish focus group 1)

The community in the support group had the potential to transform the participants’ understanding of themselves, of others and their social conditions supporting them in regaining life in the aftermath of the suicide loss.

### 3.2. An Alternative Space for Belonging and Grieving

The peer-led support group as an alternative space for belonging and grieving described how the participants felt, whereby they could give themselves permission to grieve and mourn within the group. The feeling of belonging was based on the bereavement by suicide as a common denominator, making them feel a mutual understanding, as opposed to their experiences from other settings. Gale described this in his first meeting with peers:


*Gale: I picked myself up and decided I had to go to the next ‘walk and talk’, and all of a sudden, it made a real difference because they (peers) were someone who I could talk to; they understood. It did not sound out there at all—like others had thought prior to this. ‘You just need to hit the restart button’ and ‘get back on the horse’ and whatever else people say. I even have a perfectly pleasant elderly woman for a neighbor who decided she would help me sign up for dance classes. This way I would have some positive experiences and maybe even meet a new girlfriend.*



*Ingrid: Oh no. [Sympathetic laughter in the group].*



*Gale: After a couple of months, right. [laughs himself].*



*Taylor: You simply needed to take a dance class and then everything would magically be okay again. [Sympathetic laughter in the group].*



*Ingrid: That is just how things are.*



*Gale: Exactly.*



*Ingrid: However, they do mean well.*



*Gale: And I know that.*
(Danish focus group 2)

The participants in the above interaction, by their sympathetic laughter and supportive statements, also reveal how they recognized Gale’s experience of being advised to get over the grief; while well-intended, it was out of place from an inside perspective. Participants described how they experienced an instant mutual understanding and a shared language by being with peers. In contrast to other settings, they did not have to explain or defend how they felt on a given day:


*Mary: When I came across the group, it was such an overwhelming thing to be with people. I didn’t have to explain [to them] if I was doing well or having a good day. And on another day, I didn’t have to explain that I still missed my daughter or that I am still heartbroken. And if I wanted to talk about her or not, they knew. It was like they did away with the word ‘should’, so it was complete acceptance [of members within the group].*
(Irish focus group 1)

The alternative space for belonging and grieving was created and sustained by group actions and rituals, for example, lighting a candle of confidentiality, mentioning group ground rules, cozy gatherings and a round of introductions where all participants presented themselves as bereaved by suicide. They experienced the group as a non-judgmental space, which confirmed that everything could be shared; no feelings, reactions or thoughts were wrong or unacceptable. Further, it was also a space that provided the opportunity to openly deal with the grief, which they often had to conceal in their daily lives. This ability to speak freely within the group and discuss the ‘undiscussable’ made it possible to strengthen their identity as bereaved by suicide in their ordinary lives, even though the pain of the loss was continuously present:


*Ashley: When you are in that space [the support group] where you have the opportunity to deal with your grief, where you can have your grief and you enter this space of grief. Not that you put on a facade, but you simply do not have room for your grief in a working life—or at least I do not in mine […] To me, what has been important is that you are active with the organization, and then you sort of withdraw and go back to your ‘normal’ life where your grief is a bit of a taboo. Here you need to live life the way you always have and then kind of be allowed to return to your grief. It is a way to bring… to allow myself to grieve and mourn and process my grief in this space […].*
(Danish focus group 1)

The group provided a space where people could share thoughts, feelings and challenges without fear of judgment or stigmatization, as the conventional expectations around grief in other contexts were suspended. As a result, participants felt safe to tell and re-tell their stories to strangers with different backgrounds, genders, ages, etc., without feeling the need to explain or censor themselves for fear of burdening others:


*Kate: You could cry if you wanted to, it was such a relief that you could cry and that it didn’t make other people uncomfortable. I couldn’t grieve in front of my husband, not because he isn’t wonderful, he is, but if he was having a better day, no way was I going to tell him I am having a rough day and drag him down.*
(Irish focus group 2)

They inherently trusted their thoughts, reactions and feelings to be bearable for their peers, but our findings also contained rare examples of the opposite. For example, a participant mentioned she also was grateful for the death of her beloved one, because she had learned some profound lessons about life. Another participant reacted to this statement, as perceiving the loss as something ‘appreciative’ was not something she could understand. The first participant tried to repair the conflict by saying that it was her personal experience and perhaps verbalizing or mentioning to the group may have been inappropriate:


*Max: […] I thank [name of deceased] for starting a process for me. I think there is a gift for us in what happens, regardless of what comes. It is my conviction, with reservations for all the losses and all the grief.*



*Vinnie: I haven’t exactly figured out what I should learn.*



*Ashley: It can take time to find it. [Laughs in disbelief].*



*Max: Well I don’t think of it as something to rush and there is also a difference in exactly what you lose.*



*[Silence]*



*Vinnie: Well, no matter what, it was still something they decided… I guess the relationship is a bit subordinate in my eyes, still, I don’t understand… What am I to learn from this?*



*Max: It may not have been appropriately said by me.*



*Vinnie: Well, I don’t know.*



*Max: It is just what I experience in relation to myself.*



*Vinnie: Yes, and I am not saying that it is wrong. I’m just saying I haven’t found that common thread.*



*Max: No no.*



*Vinnie: I don’t know if I ever will.*
(Danish focus group 1)

Later, group members acknowledged they also thought they had acquired new perspectives, but highlighted that it had come at a high price.

The peer-led support group was described as a unique space where the participants not only had the possibility to focus on themselves as bereaved by suicide but were also allowed to disclose how they felt. This was an invaluable alternative to other contexts in their daily living.

### 3.3. A Conflicted Space

Some participants experienced participating and belonging to the group as troublesome. They deeply appreciated and needed the connectedness and shared experience, but at the same time felt discomfort in the encounter with peers, which was also a hardship. For example, the participants had never desired to be a part of the bereavement group. It was not their choice to become bereaved by suicide. The experience created a feeling of being distant from people in their usual social network, which led them to seek connections with peers. Meeting peers was described as both pleasant and unpleasant. On the one hand, the peer community was perceived as a valuable and helpful space where they felt comforted and could express their true feelings about the tragedy. On the other hand, it was a space that they did not want to belong to:


*Paul: I think for me the main thing of the group is there’s a sort of, it’s a difficult thing to say because it is a group, you don’t really, you wish you weren’t in, but there’s a comfort in it, when you meet people and you’re talking to people. There’s a comfort in it because you feel you can sort of relax and you can say you know what you want.*
(Irish focus group 1)

However, being confronted with other people’s grief could also be discomforting, as this activated reflections on their own grief. They felt it was necessary to enter the space of the support group, but they were occasionally tempted to avoid coming to meetings, as described by Ashley:


*Ashley: Leading up to a meeting, I could ask myself if maybe I was coming down with something [laugh] or the like. Because you need to walk in and out of that room of grief, and sometimes it is simply easier just not to walk in in the first place. Coming up with reasons not to go is easy, but you… you need to be able to take part in it […]. Otherwise, I might stay on the surface and make sure everything else runs smoothly, without actually noticing myself all that much. So, it is good to have a space where I have to take stock of… of me.*
(Danish focus group 1)

Indeed, some participants spoke of how they had to take a break from the group for shorter or longer periods because the stories from peers at times could be too hard to take in:


*Clare: That was actually one of the reasons I stopped joining the group’s sessions; it began to take up too much and my cup ran over almost. Both with other people’s grief… and it was… it was not that I did not gain anything from hearing about how others deal with matters and such, but it also started things in me that was hard to handle.*
(Danish focus group, 2)

The fact that group membership was not time-limited and that people had the freedom to step out for periods of time, helped participants feel they were in control. It was also good to know that the other participants fully understood if one was unable to participate because of obligations in their daily living or lack of energy. This freedom also ensured that participants actively chose to step into the grief when they were present in the peer-led support group. In addition, knowing that they could always return to a welcoming group without explanation or justification was viewed as an important lifeline.

The determination to participate despite the discomfort and hardship was driven by the valuable contribution the peer group made. Participating in the peer-led support group can be understood as a strong driving force towards making room for and learning to live with the new conditions of life and reclaiming viability in daily living, despite the conflicts embedded in the space.

## 4. Discussion

In this study, those bereaved by suicide described their experiences of being part of a peer-led support group and how it affected them, mostly in contrast to encounters with non-bereaved individuals, both laypeople and professionals. The findings do not imply that all professionals were unhelpful; however, they do emphasize the particularity and strength of being with peers. Despite being a conflicted space for some, the peer-led support group provided a distinct and legitimate space for grieving and belonging and a transformative space supporting learning, change and finding ways to process their grief and handle the expectations of people in social interactions in everyday life. Although the themes were presented separately, in reality, they are interconnected, as the experience of the group as ‘an alternative space for belonging and grieving’ was essential for the experience of the peer-led group as ‘a transformative space’. The experience of gaining and learning through the group also made it worth overcoming feelings that it was a conflictual space to participate in.

Participants not only had to deal with the loss of a loved one; being bereaved by suicide created a rupture in their sense of belonging with their social network within their community. They felt alienated, as their grief did not fit with the expectations of their surroundings. As summarized in a systematic review [4], those bereaved by suicide often report feelings of shame and guilt, which complicates the process of grieving and affects their ability to grieve openly. In order to avoid negative reactions, some bereaved resorted to isolation and concealed their thoughts and emotions. The awkwardness, discomfort and embarrassment experienced in social settings resulted in inadequate social support, and the bereaved found themselves alone in their grief [4]. In this study, participants reported a strong consensus about the ‘us’ and ‘others’ as they perceived especially non-bereaved laypeople and some professionals as being unable to understand their profound grief as a new condition of life. According to the theory of belonging, human beings have a universal basic need to form and maintain at least a minimum quantity of interpersonal relationships and that people are naturally driven toward establishing and sustaining a state of belonging [35]. We found that participants perceived a special connectedness, sense of belonging and mutual understanding in the non-judgmental environment of the peer-led support group. This aligns with previous findings that loneliness and stigma from social networks and community encouraged those bereaved by suicide to search for a safe, understanding and non-judgmental environment with peers [21]. The participants’ search for a peer community may be viewed as a means to compensate for the lack of support in their social network. This is in line with the study by Griffin et al. [36] that linked improved wellbeing and a reduction in somatic grief reactions after community-based peer support to the participant’s descriptions of the group as a safe space to talk and that provides a sense of belonging and connection.

Concealable stigmatized identities are aspects of the self or characteristics that are socially devalued with negative stereotypes, which can be hidden for others, such as bereavement by suicide. People with stigmatized identities often use concealment in order to increase social belonging and reduce personal experiences of discrimination. However, active concealment can also hinder their feeling of connectedness and well-being [37]. Levi-Belz and Lev-Ari [38,39] found that self-disclosure among suicide-bereaved individuals increased feelings of belonging and perceived social support, which in turn reduced complicated grief and facilitated post-traumatic growth and recovery. The findings of the current study indicated that it was important for the participants to be allowed to grieve in their own way when in the peer-led support group. People with concealable stigmatized identities have to determine in which contexts they conceal or disclose, both having benefits and disadvantages, as it is likely to impact the social interactions, sense of belonging and physical and psychological well-being [37]. Our findings indicated that the participants often preferred to withhold disclosure as it helped them focus on usual activities in daily living and live up to the expectations of their social network. However, being in the support group also empowered them to share their grief with relatives or non-bereaved outside the group, thus reducing the feeling of alienation. This is in line with previous findings linking peer-led suicide bereavement interventions to reduced feelings of stigma [21]. However, the participants in our study described daily living wherein they often concealed their true state of mind from people in their surroundings, due to adverse reactions. Disclosing personal and emotional information can benefit the grieving process [40]. The participants primarily used the peer-led support group to disclose their grief and described it as helpful, as they felt they became better at handling and adjusting to the expectations of their wider community. Thus, they did not necessarily explore opportunities for support in their social network; some participants resigned to the fact that non-bereaved people would not understand and only revealed their true state of mind when in the peer support group. Concealment may protect from occasions of prejudice and discrimination, but it can also limit social support. Disclosure can increase social support, but disclosing to unsupportive others may also lead to psychological distress [37]. Our findings indicate that although the participants found social support inside the group, some participants found the group had enabled them to talk more openly to others outside, but they still longed for more attention and support in their social network.

### Limitations and Strengths

This study has some limitations: the two organizations differed in the kind of activities they offered and had different organizational structures. FOSL is governed by a central board, whereas NABS consists of a nationwide umbrella organization for many local organizations that are governed by local volunteers. Although this complicated our comparisons, the analysis revealed similarities in the associations’ core principles and in the experiences from the participants.

In addition to recall bias, social desirability bias may have also been an issue, as data were collected by focus groups. Given the sensitive nature of the research topic area, participants may have consciously or unconsciously concealed some of their impressions. Further, the focus groups were conducted online and face to face, respectively, in the Republic of Ireland and Denmark. In FOSL, the participants were already using online support groups as they had adjusted to the COVID-19 situation. Therefore, representatives from FOSL deemed it possible to carry out the focus groups online as well. In the Danish setting, the representatives from NABS stressed that the focus groups should be face to face, as their members had no experience in online meetings. The different formats might have influenced the findings. Further, the relatively small sample size and the fact that participants were self-selecting may also have biased the findings. All participants were positive and grateful users of a peer-led organization, implying that experiences of those who chose to leave the group were not represented. Thus, negative or challenging aspects of participating in a peer-support group, such as the pain of listening to people’s stories, feelings of being burdensome or re-traumatized, and the lack of a group facilitator [41,42], may be under-represented. Still, the findings revealed that participating in a peer-led support group could evoke conflicts, suggesting a need for further research into people who do not engage with or have left the peer support group would be useful for both organizations. There is also a need for further research investigating how stigma can be reduced outside the peer-led support groups. The overall findings reflected the experiences of people within two specific communities and may not apply to or fit other settings. However, the cross-cultural similarities related to participation and experienced effects of the peer-led support group were underlined, as the findings correlated with the findings from a scoping review including studies from five additional countries (USA, Canada, England, Italy and Sweden) [21].

It was a strength that the participants were purposefully sampled to represent people who engaged in peer-led support groups, and thus they had experiences with the subject under investigation and could therefore share how they perceived participating in peer-led support groups and how it affected them. Another strength was that the trustworthiness of the qualitative, descriptive research’s version of rigor addresses credibility, confirmability, dependability, transferability and reflexivity applied [25,43]. Table 2 provides a summary of the provisions made by the researchers in this study to address these issues.

## 5. Conclusions

Despite being placed in two different cultural settings, i.e., Denmark and the Republic of Ireland, and different organizational approaches, comparable experiences were reported by people bereaved by suicide in the peer-led support groups. Although peer-led support groups may not be helpful for all bereaved by suicide, they can provide supportive spaces that aid the participants’ recovery process. In terms of facilitating recovery, peer-led support groups may aid people bereaved by suicide process the grief and cope with societal expectations and act as a vehicle to help circumvent the problem of stigma surrounding suicide.

## Figures and Tables

**Table 1 ijerph-19-09898-t001:** The participants.

Participant	Male	Female	Loss of Child	Loss of Spouse/Partner	Loss of Sibling	Multiple Loss *
Denmark (*n* = 14)	4	10	7	5	2	4
Ireland (*n* = 13)	4	9	7	3	3	1
In all (*n* = 27)	8	19	14	8	5	5

* Multiple loss is in addition to the primary loss.

**Table 2 ijerph-19-09898-t002:** Trustworthiness of data.

Quality Criterion	Provisions Made by Research Team
Credibility is the confidence in the findings	The focus group schedule was developed in collaboration between researchers and experts by experience Participants were heterogeneous (gender, duration of contact with the groups, different kinship to the deceased, diverse cultures and backgrounds) The moderator had contact with each of the participants before the focus groups and initiated a trusting relationship Data were systematically compared across groups and countries
Confirmability is the extent to which the findings of a study are shaped by the respondents and not researcher bias, motivation or interest	Analysis was completed by more than one person and interpretations were agreed upon collectively Codes and themes were discussed between researchers to minimize interpretative bias Presented data extracts included examples of exceptions/variations of the themes Lengthy quotes were used to demonstrate and support interpretations
Dependability is the description of the conduct of the study, including changes to make procedures and processes clear	The description of the study’s methods to enable replication Detailed descriptions of the coding process included the changing perspectives in the interpretive process
Transferability is about the degree to which findings have applicability in other contexts by making the context explicit and descriptive	Purposive sampling included people bereaved by suicide that had participated in peer-led support groups Descriptions of research setting, peer-led interventions and participant profiles were provided Data extracts included rich descriptions of the themes with direct quotes
Reflexivity is critical self-reflection of how the researchers’ social background, assumptions, positioning and behavior may have affected the research process	Care was taken to work reflexively by continually questioning interpretations, assumptions and positions The analytical process was based on an ongoing dialogue between members of the research team who had different background and expertise

## Data Availability

The data are not publicly available due to the rules of the ethical committee.

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
