# Peer review of "The Spaces of Peer-Led Support Groups for Suicide Bereaved in Denmark and the Republic of Ireland: A Focus Group Study"

_ijerph, 2022, doi:10.3390/ijerph19169898_

Round 1

Reviewer 1 Report

The manuscript is very well-written and highly interesting. I have only some questions to consider or minor recommendations for the authors, listed by lines for clarity: 

Line 60: I suggest writing “out of which 35% indicated …” instead of “and” to increase clarity

Line 103: Is it “analogous assessment« or »analogue assessment”?

Line 158: Is it “were contacted by the research team” or “contacted the research team”?

Line 167: I suggest changing “multiple suicides” to “multiple losses by suicide”

Line 170: Is it necessary to mention “members of the same family”? As I am very cautious about noting any potentially revealing information I am thinking about how to make this vaguer, or not to mention this information at all.

Lines 191 and 192: “Focus groups were audio recorded”; I suggest explicitly stating, that all (both face to face and online) focus groups were audio recorded, as well as the interviews.

Lines 224 and 225: I suggest starting the sentence without “although” and adding “however, these were merged …”

Line 244: If I understand correctly, the whole coding process took place for each country; therefore the first sentence of this paragraph seems to be redundant.

Line 265: Perhaps a statement regarding the (non) sharing of raw data outwards (i.e. open science) can be made additionally at this point.

Lines 302-305: At the first glance the example presented here seems like an example for describing the peer-led support group as “an alternative space”, that is, the second theme identified in the study. However, it seems to me that these two themes are quite interdependent. That is since the group is experienced as “an alternative space” this can be a prerequisite of the way it can also be then experienced as “a transformative space”. Perhaps this interrelatedness of the themes could be discussed shortly in the paper?

Relating to all participants’ quotes: Is it necessary to specify the group of the individual participant (e.g. Danish focus group 1 etc.)?

Lines 341 and 342: Full stops are missing at the end of the sentences

Line 348: Is there something missing in the sentence? (“… these would not be able …”); e.g. “these people” or “they”

Lines 406 and 407: Is there something missing in the sentence? (“… like they did away with the word …”) I am not sure if this is due to the informal way of speaking, though.

Line 442: Did she say that it was inappropriate that “she mentioned it” or perhaps “the way she mentioned it”? It would be interesting if you can add any further thoughts about this situation since it seems like an emotionally charged and relevant event. As described in the paper earlier, the group dynamics of the peer-led support groups were reflected in the group dynamics of the focus groups. It may be interesting to expand on this particular example – if the data seems relevant and not potentially revealing at the same time. This is just my impression as a reader.

Lines 500 and 501: I suggest perhaps adding “mostly in contrast …” My further thoughts on this: Do you have any examples of contrasts of experiences with professionals? These are only mentioned at the beginning of the Results section, but no examples are described in further detail within the three main identified themes. It would make sense that most described contrasts relate to the daily encounters with the bereaved individuals’ social network lay people – however, if this is the case, perhaps a distinction, even if subtle, between lay people and professionals should be made. It is nicely described in the paper, that the bereaved did not find a sense of community or togetherness with anyone else, that is lay people or professionals (whether they found professionals helpful otherwise or not). However, from the clinical practice point of view, it makes a difference if the bereaved also felt that they cannot fully narrate their stories or disclose their emotions to the professionals, for example, due to the fear of burdening them as described in one example of participant’s relationship with her husband. Even though this is not the central topic of the paper, mentioning details like this may make a subtle difference when reading it.

Line 510: It seems like “affected” should be changed to “affects”.

Lines 511-514: The findings described here could potentially be nicely supplemented by an additional sentence or two about findings with regards to the bereaved relationships with professionals (as discussed above: do they also conceal their thoughts and emotion with them; perhaps not, but it may be questionable, how often they even seek/receive professional help?). Again, I realize that this is not the paper’s central topic, but it could make a nice addition.

Line 527: I suggest adding that there was a reduction in “somatic” grief reactions, for clarity.

Line 549: Perhaps some details of the cited study are not necessary at this point (e.g. the number of participants); also – the findings of this study (no. 40) seem to be replicating those mentioned earlier (no. 38 and 39). Could these be merged a bit?

Lines 555-557 and 561: … However, some participants did find it helpful to also talk openly with people in their social network, with the group being seen as crucial support for them to be able to do so – this is mentioned in the results section, I suggest it to also be discussed at this point a bit.

Lines 575-577: The given information may be easier to grasp for the reader with small additions: “the participants were already used to online support groups …” and “… to carry out the focus groups online as well.”

Table 2:

-          “Diverse groups of participants were included”; I believe this would be more accurate formulated like this: “Participants were heterogeneous”

-          “tables made public” are mentioned; it seems a bit unclear, which tables are referred to  

-          “Presented data extract include examples exceptions/variations …”; should it be “extracts”? and “examples of exceptions/variations …”?

Author Response

Thank you for the very helpful review. Please see the attachment for our point-by-point response. 

Reviewer 2 Report

This is a good quality manuscript that I have enjoyed reviewing. I have made my comments in the attached word document. I wish you well as you progress toward publication with this work.

Author Response

Thanks for your very helpful review. Please see the attachment for our point-by-point response. 
